# Aerobic Exercise Prevents Arterial Stiffness and Attenuates Hyperexcitation of Sympathetic Nerves in Perivascular Adipose Tissue of Mice after Transverse Aortic Constriction

**DOI:** 10.3390/ijms231911189

**Published:** 2022-09-23

**Authors:** Niujin Shi, Jingbo Xia, Chaoge Wang, Jie Zhou, Junhao Huang, Min Hu, Jingwen Liao

**Affiliations:** 1Guangdong Provincial Key Laboratory of Physical Activity and Health Promotion, Guangzhou Sport University, Guangzhou 510500, China; 2Graduate School, Guangzhou Sport University, Guangzhou 510500, China; 3Key Laboratory of Regenerative Medicine, Ministry of Education, Jinan University, Guangzhou 510500, China; 4School of Kinesiology, Shanghai University of Sport, Shanghai 200438, China; 5Scientific Research Center, Guangzhou Sport University, Guangzhou 510500, China

**Keywords:** arterial stiffness, sympathetic nerves, PVAT, exercise, heart failure

## Abstract

We aimed to investigate the efficacy of exercise on preventing arterial stiffness and the potential role of sympathetic nerves within perivascular adipose tissue (PVAT) in pressure-overload-induced heart failure (HF) mice. Eight-week-old male mice were subjected to sham operation (SHAM), transverse aortic constriction-sedentary (TAC-SE), and transverse aortic constriction-exercise (TAC-EX) groups. Six weeks of aerobic exercise training was performed using a treadmill. Arterial stiffness was determined by measuring the elastic modulus. The elastic and collagen fibers of the aorta and sympathetic nerve distribution in PVAT were observed. Circulating noradrenaline (NE), expressions of β3-adrenergic receptor (β3-AR), and adiponectin in PVAT were quantified. During the recovery of cardiac function by aerobic exercise, thoracic aortic collagen elastic modulus (CEM) and collagen fibers were significantly decreased (*p* < 0.05, TAC-SE vs. TAC-EX), and elastin elastic modulus (EEM) was significantly increased (*p* < 0.05, TAC-SE vs. TAC-EX). Circulating NE and sympathetic nerve distribution in PVAT were significantly decreased (*p* < 0.05, TAC-SE vs. TAC-EX). The expression of β3-AR was significantly reduced (*p* < 0.05, TAC-SE vs. TAC-EX), and adiponectin was significantly increased (*p* < 0.05, TAC-SE vs. TAC-EX) in PVAT. Regular aerobic exercise can effectively prevent arterial stiffness and extracellular matrix (ECM) remodeling in the developmental course of HF, during which sympathetic innervation and adiponectin within PVAT might be strongly implicated.

## 1. Introduction

Arterial stiffness is an independent predictor of cardiovascular events [1], among which heart failure (HF) can be one of the most serious outcomes [2]. The development of arterial stiffness is typically attributed to extracellular matrix (ECM) remodeling within media and adventitia [3], represented by the fragmentation of elastic fibers and deposition of collagen fibers [4]. Recently, perivascular adipose tissue (PVAT), an important cross-linking connective tissue, was highlighted as a mediator of ECM remodeling [5]. PVAT surrounds most of the vasculature and has emerged as an active component of the blood vessel wall [6], regulating vascular homeostasis [7] and affecting the pathogenesis of arterial remodeling [6]. Pathological processes including HF [8] and obesity [9] usually drive hyperexcitation of sympathetic nerves. Sympathetic innervation within PVAT has recently been recognized as a possible regulator of ECM remodeling during the development of arterial stiffness [10]. Previously, the release of adiponectin via activation of β3-adrenergic receptor (β3-AR) in PVAT was reported to be triggered by sympathetic stimulation [11]. Adiponectin, as one of PVAT-derived adipokines, was implicated in arterial stiffness [12] and had beneficial effects in maintaining vascular homeostasis [13].

A meta-analysis of randomized trials supports that exercise-based cardiac rehabilitation should be offered to all HF patients [14], and quantitative evidence from patients with cardiovascular disease also suggests that exercise interventions including aerobic, combined, or isometric trainings are capable of reducing arterial stiffness [15]. Moreover, exercise training would normalize the hyperexcitation of sympathetic nerves in HF [16]. However, the efficacy of exercise on preventing arterial stiffness and the potential role of sympathetic nerves within PVAT are not fully understood.

Therefore, this study was designed to determine the effects of 6 weeks of aerobic exercise training on preventing arterial stiffness in pressure overload-induced HF mice and to explore the potential roles of sympathetic nerves within PVAT. We hypothesized that 6 weeks of aerobic exercise training would be effective in preventing arterial stiffness and ECM remodeling and attenuating the hyperexcitation of sympathetic nerves in PVAT during the developmental course of HF.

## 2. Results

### 2.1. The Developmental Course of Heart Failure after TAC Was Delayed by Exercise

Both the ejection fraction (EF) (Figure 1A) and fractional shortening (FS) (Figure 1B) of TAC-SE were significantly reduced (*p* < 0.05) compared with those of SHAM; this reduction was significantly restored (*p* < 0.05) in TAC-EX. The heart weight (HW) normalized to body weight (BW) (Figure 1C) and HW normalized to tibia length (TL) (Figure 1D) of TAC-SE were significantly elevated (*p* < 0.05) compared with those of SHAM, whereas HW/BW and HW/TL of TAC-EX were significantly reduced (*p* < 0.05) compared with those of TAC-SE. The representative wheat germ agglutinin (WGA) staining images of cardiomyocyte (CM) is shown in Figure 1E; the CM size of TAC-SE was significantly increased (*p* < 0.05) compared with that of SHAM, and the CM size of TAC-EX was significantly decreased (*p* < 0.05) compared with that of TAC-SE (Figure 1F).

### 2.2. The Remodeling of ECM after TAC Was Prevented by Exercise

EVG staining of the thoracic aorta (Figure 2A) showed that the elastic fibers area of TAC-SE was decreased compared with that of SHAM, and the elastic fibers area of TAC-EX had no statistical difference compared with that of TAC-SE (Figure 2B). Sirius red staining of the thoracic aorta was applied to identify the collagen fibers area (Figure 2C). Compared with SHAM, the collagen fibers area of TAC-SE was significantly larger (*p* < 0.05); compared with TAC-SE, the collagen fibers of TAC-EX were significantly decreased (*p* < 0.05) (Figure 2D). H&E staining of the thoracic aorta (Figure 2E) showed that the aortic wall thickness/diameter of TAC-SE was significantly increased (*p* < 0.05) compared with that of SHAM, and the thickness/diameter of TAC-EX was significantly decreased (*p* < 0.05) compared with that of TAC-SE (Figure 2F).

### 2.3. The Thoracic Arterial Stiffness after TAC Were Prevented by Exercise

A representative image of aortic elastic modulus tests of each group is shown in Figure 3A. The elastin elastic modulus (EEM) of TAC-SE was significantly decreased (*p* < 0.05) compared with that of SHAM, and the EEM of TAC-EX was significantly increased (*p* < 0.05) compared with that of TAC-SE (Figure 3B). The collagen elastic modulus (CEM) of TAC-SE presented as significantly higher than that of SHAM (*p* < 0.05), and the CEM of TAC-EX displayed a significant decrease (*p* < 0.05) compared with that of TAC-SE (Figure 3C). 

### 2.4. Hyperexcitation of Sympathetic Nerves within PVAT after TAC Was Attenuated by Exercise

The circulating noradrenaline (NE) of TAC-SE was significantly higher (*p* < 0.05) than that of SHAM; the NE of TAC-EX was significantly restored compared with that of TAC-SE (*p* < 0.05) (Figure 4A). The thoracic aortic PVAT was further stained with tyrosine hydroxylase (TH) (a marker of sympathetic neurons) to evaluate the distribution density of sympathetic nerves (Figure 4B,C); the results showed that the sympathetic nerve density of TAC-SE was significantly higher (*p* < 0.05) than that of SHAM, and the sympathetic nerve density of TAC-EX was significantly lower (*p* < 0.05) compared with that of TAC-SE. The BW and thoracic aortic PVAT of mice were weighed. The PVAT (mg) and PVAT (mg)/BW (g) were significantly increased (*p* < 0.05) after exercise compared with those of the TAC-SE group (Table 1).

### 2.5. β3-Adrenergic Receptor (β3-AR) and Adiponectin within PVAT after TAC Was Influenced by Exercise

The expression of β3-AR in TAC-SE was significantly increased (*p* < 0.05) compared with that of SHAM, and that in TAC-EX was significantly decreased (*p* < 0.05) compared with that of TAC-SE (Figure 5A). There was no significant difference in the expression of adiponectin in PVAT between SHAM and TAC-SE, and a more significant increase (*p* < 0.05) was found in TAC-EX than in TAC-SE (Figure 5B).

## 3. Discussion

In this study, we examined the efficacy of 6 weeks of aerobic exercise training on preventing arterial stiffness in HF and the contribution of the sympathetic nerve within PVAT to this process. These are the primary findings: (1) aerobic exercise training significantly prevented arterial stiffness represented by ECM remodeling and an elastic modulus shift during HF; (2) sympathetic activation in PVAT may have contributed to the reconstruction of ECM, during which expressions of adiponectin in PVAT significantly changed. These findings provide novel evidence of the effects of aerobic exercise on preventing arterial stiffness during HF and suggest a potential role of sympathetic innervation of PVAT in this process.

Regular exercise prevented arterial stiffening and ECM remodeling. The effects of exercise training on HF was systematically reviewed [17] and the findings supported our results that exercise recovered cardiomyocyte morphology and cardiac ejection fraction. A primary finding of the present study is that aerobic exercise training prevented arterial stiffness during the development of pressure overload-induced HF. Regular aerobic exercise, regardless of intensity or duration, is effective in reducing arterial stiffness [18], which is a critical contributor to HF [19]. The mechanical stiffness of the aorta is dependent on elastin and collagen, which are prominent scaffolding proteins of ECM within the vascular wall [3]. It is generally considered that the elasticity of arteries is facilitated by elastin and stiffness by collagen [3]. Based on the observations of thoracic aortic morphology, our data indicated that the development of HF was accompanied by increased arterial collagen fibers and decreased elastin fibers, whereas aerobic exercise offset those influences. Notably, Ouyang et al. already found that both long-term chronic continuous and interval exercise training could prevent ECM remodeling of the coronary artery in pressure-overload-induced HF mice [20]. We additionally showed that thoracic aortic ECM remodeling and arterial stiffness were prevented by regular aerobic exercise in pressure overload-induced HF.

Sympathetic innervation within PVAT was recently recognized as a possible regulator of ECM remodeling during the development of arterial stiffness [11]; however, this effect has not been confirmed in the setting of exercise intervention. Our results showed that aerobic exercise attenuates the hyperexcitation of sympathetic nerves in PVAT, represented by the reduced expression of circulating NE and the distribution of the sympathetic nerve. Exercise training has been shown to normalize sympathetic hyperexcitation in patients with HF [16], and to restore PVAT function and prevent vascular complications [21]. Quantitative evidence from HF patients systematically supported this positive influence of exercise [22]. 

The sympathetic nerve of PVAT could derive NE and activate β3-AR [11]. To the best of our knowledge, the influence of exercise on β3-AR expression was only reported in obesity [21]; β3-AR expression in mesenteric PVAT was downregulated in obesity but improved with exercise [21]. In this study focusing on HF, we observed increased β3-AR expression in aorta PVAT, and aerobic exercise restored its expression. This inconsistence may be explained by difference pathological conditions. A previous study also suggested that β3-AR might undergo desensitization due to sustained sympathetic adrenergic activation in HF [23]. In addition, a recent study also highlighted a role of organic cation transporter-3 on absorbing excessive NE in PVAT, thus preventing β3-AR overactivation by NE [11]. 

We previously hypothesized that β3-AR mediated adiponectin release in PVAT, because a reduced expression of β3-AR was demonstrated to result in a reduction in adiponectin secretion from mesenteric PVAT [21]. Our results showed that aerobic exercise restored the expression of adiponectin from PVAT, even though the expression of β3-AR decreased. This may be due to the secretion of adiponectin in PVAT being influenced by other factors in addition to β3-AR, including the glucocorticoid receptor in the adipose tissue of obesity [24] and 4-hydroxynonenal (a product of lipid peroxidation) in the PVAT of atherosclerotic patients [25]. The protective role of adiponectin in other conditions were also restored by exercise training: aerobic exercise increased adiponectin expression in the aortic PVAT of type 2 diabetic mice [26]; similar results were also found in the aortic PVAT of Zucker rat under chronic stress [27]. Therefore, our results suggested that aerobic exercise restores adiponectin in PVAT and prevents arterial stiffness during HF development.

The limitation of the present study is that we only quantified the expression of β3-AR in PVAT but did not evaluate the desensitized level of β3-AR or other potential factors involved in regulating adiponectin; therefore, the mechanism of β3-AR-mediated adiponectin release could not be depicted.

In summary, aerobic exercise prevents arterial stiffness during the development of HF, and sympathetic nerve innervation and adiponectin within PVAT are strongly implicated in this process. Our study highlighted PVAT sympathetic innervation as a promising aspect for the prevention of arterial stiffness in HF.

## 4. Materials and Methods

### 4.1. Animal Care and Experimental Design

All animal care and experimental procedures were approved by the Animal Experimental Ethics Inspection of Guangzhou Sport University and performed in accordance with the principles of the Declaration of Helsinki. A total of 53 male C57BL/6 mice (aged 8–9 weeks) were obtained from the Animal Center of Guangdong. All animals were housed at 24 ℃ and 55% humidity under a 12 h alternating light and dark environment and fed with ordinary standard chow throughout the study. Mice were randomly divided into sham operation (SHAM) (*n* = 15), transverse aortic constriction-sedentary (TAC-SE) (*n* = 13), and transverse aortic constriction-exercise (TAC-EX) (*n* = 17) groups.

### 4.2. Establishment of Transverse Aortic Constriction Model

Transverse aortic constriction (TAC) is a commonly used experimental model for pressure-overload-induced HF [28,29]. Briefly, mice were anesthetized with 2% isoflurane, and a left upper thoracotomy was performed with the pectoralis and intercostal muscles being dissected. Then, a nylon suture was placed around the transverse aorta and loosely tied in a knot. A presterilized 27G blunt needle was then placed into the knot; after the knot was tightened, the needle was withdrawn. Finally, the muscle and skin were sutured. SHAM mice were subjected to the same procedure but without the knot [30].

### 4.3. Exercise Protocol

Treadmill exercise intervention was performed for two weeks after TAC. Adaptive exercise intervention for 1 week was set at: 8 m/min, 1 h/day, 5 days/week, and slope = 0°; formal exercise intervention for 5 weeks was set at: 12 m/min, 1 h/day, 5 days/week, and slope = 0°. The exercise intervention was performed between 19:30 and 20:30 Animals were allowed to break three times for 3–5 min during each exercise intervention [31,32].

After the exercise intervention, cardiac function was evaluated by echocardiography. Blood samples, heart, thoracic aorta, and PVAT were collected from the mice. The thoracic aortas and hearts were fixed in 4% paraformaldehyde for histological assessments, and thoracic aortic PVAT was stored at −80 °C for western blotting.

### 4.4. Echocardiography

Echocardiography was performed after exercise using a Vevo2100 system (Visual Sonics Inc., Toronto, ON, Canada) with a 40 MHz MS550D ultrasound transducer. Mice were anesthetized with 2% isoflurane and were depilated with depilatory cream, and then ultrasonic coupling agent was applied. Heart and respiration rates were continuously monitored via stage electrodes. Scanning was initiated with parasternal long- and short-axis views [33,34]. End-diastolic volume (EDV) and end-systolic volume (ESV) were calculated using the Simpson method of disks [35]. The ejection fraction (EF) was calculated: EF (%) = (EDV − ESV)/EDV × 100. The left ventricular end-diastolic dimension (LVDd) and left ventricular end-contractile diameter (LVDs) were measured in one-dimensional mode. Fractional shortening (FS) was calculated: FS (%) = (LVDd − LVDs)/LVDd × 100.

### 4.5. Histological Assessments

The left ventricle of the heart was collected and fixed with 4% paraformaldehyde, embedded in paraffin, and cut with a thickness of 5 μM. Sections of the heart were stained with WGA kits (Thermo-Fisher, Waltham, MA, USA) to measure myocardial cell size [36]. A cross-section of cardiomyocytes was photographed under a fluorescence microscope, and 1000 cells were counted in each section for analysis.

The thoracic aorta was collected and fixed with 4% paraformaldehyde. The tissue was embedded in paraffin and cut to a thickness of 5 μM. For morphometric analysis, sections of the thoracic aorta were stained with hematoxylin and eosin (H&E) (Bio sharp, Hefei, China) and imaged using an Olympus BX53^®^ (Olympus, Center Valley, PA, USA). The wall thickness and inner diameter of aorta were analyzed using ImageJ software (ImageJ 1.53e, National Institution of Health, Montgomery, MD, USA). For ECM analysis, sections were stained with Sirius Red (Bio sharp, Hefei, China) for collagen fiber distribution and with Elastica-Van Gieson (EVG) staining (Bio sharp, Hefei, China) for elastic fiber distribution [37]. The area of elastic fiber was calculated as the elastic fiber area/cross-sectional area of aorta ×100%, and the expression of collagen fiber was calculated as collagen fiber area/cross-sectional area of aorta ×100%. To observe the sympathetic nerve distribution within PVAT, sections were stained with tyrosine hydroxylase through immunohistochemistry. Sections were then incubated with TH antibody (abs131679, absin, Shanghai, China,) at 4 °C overnight [38,39]. Imaging was captured using a microscope at 1000× magnifications and analyzed using ImageJ software (ImageJ 1.53e, National Institution of Health, Montgomery, MD, USA).

### 4.6. Mechanical Stiffness Testing

Arterial elasticity and stiffness were assessed by mechanical stiffness testing [40]. Preheated 37 °C Ca^2+^ and Mg^2+^-free phosphate-buffered saline solution was added into the chamber of a Multi Wire Myograph System (Model 620M, DMT, Aarhus, Denmark), and then the thoracic aorta with no PVAT was cut into a ~2.0 mm arterial ring. The arterial ring was mounted in the chamber and was stretched 1 mm every 3 min until mechanical failure. The elastic modulus was calculated from the stress–strain curve. Strain (*λ*) = Δ*d/d*(*i*) and stress (*t*) = *λF*/2*HD*, where *F* is the one-dimensional load applied, *H* is the sample wall thickness, *D* is the length of the vessel, Δ*d* is the change in diameter, and *d*(*i*) is the initial diameter. The thoracic aortic diameter and wall thickness were assessed by H&E staining, and the vessel length was measured with a fine ruler. The elastin region, coinciding with the EEM, was calculated by the slope of the stress–strain curves at the front smooth section linear. The collagen region, coinciding with the CEM was calculated by the slope of the stress–stain curves at the tail steep linear section. The slope of the stress–stain curves was determined using GraphPad Prism (GraphPad Prism 8.02, San Diego, CA, USA).

### 4.7. Enzyme Linked Immunosorbent Assay (ELISA)

The concentration of NE in plasma was determined by ELISA kits (MM-0876M1, MEIMIAN, Hangzhou, China) according to the manufacturer’s protocol [41]. Blood samples were centrifuged to separate plasma. All samples and standards were measured in duplicate, and the optical density of the zero standard was subtracted from each value. Standard curves were fitted using nonlinear regression analysis.

### 4.8. Western Blotting

The expressions of β3-AR (61033, Sigma, St. Louis, MO, USA), adiponectin (DF7000, Affinity Biosciences, Hangzhou, China), and GAPDH (G9545, Sigma, St. Louis, USA) were quantified by Western blotting. Briefly, the thoracic aortic PVAT was homogenized in 1 mL of lysis buffer by a grinding machine (Servicebio, Wuhan, China), containing 1× RIPA (Beyotime, Shanghai, China), protease inhibitors (Beyotime, Shanghai, China), and phosphatase inhibitors (Cwbio, Beijing, China). The proteins were separated by 10% SDS polyacrylamide gels and transferred to 0.2 µM Immobilon-PSQ PVDF membranes (Millipore-Sigma, Burlington, VT, USA). The membrane was then blocked in 5% BSA for 1.5 h. The primary antibody was incubated at 4℃ overnight, and goat anti-rabbit IgG (511203, Zenbio, Chengdu, China) was used as the secondary antibody incubated for one hour at room temperature. Imaging was captured using a TANON-5200Multi imaging system (Ewell, Guangzhou, China) and analyzed using ImageJ software (ImageJ 1.53e, National Institution of Health, Montgomery, MD, USA). Quantitative data were normalized using endogenous GAPDH.

### 4.9. Statistical Analysis

Data are presented as the mean ± standard deviation (SD). Statistical significance between groups was analyzed using one-way ANOVA, and *p < 0.05* was considered as statistically significant. All data were analyzed using the GraphPad Prism (GraphPad Prism 8.02, San Diego, CA, USA).

## Figures and Tables

**Figure 1 ijms-23-11189-f001:**
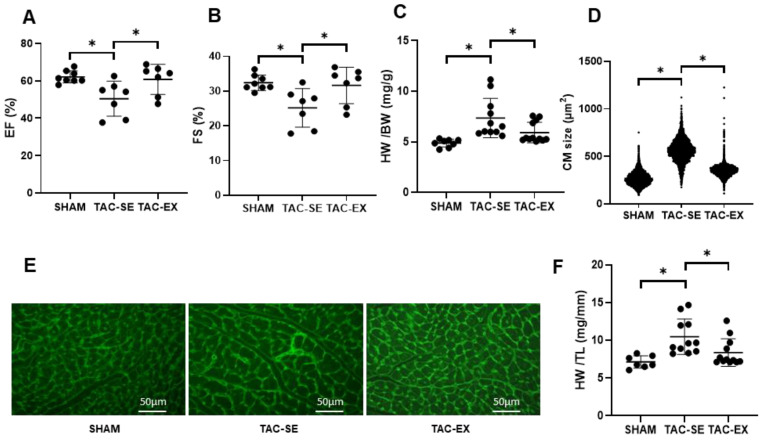
Effects of exercise on cardiac function in mice after TAC. EF (**A**) was calculated by using the formula (EDV - ESV)/EDV × 100%, and FS (**B**) was calculated by using the formula (LVDd − LVDs)/LVDd × 100%. Heart weight per body weight (HW/BW) ratio (**C**) and heart weight per tibia length (HW/TL) ratio (**D**). Original image following WGA staining (**E**), and CM size was measured (**F**). Data were analyzed using one-way ANOVA; values are mean ± SD. * indicates *p* < 0.05. Abbreviations: EF, ejection fraction; EDV, end-diastolic volume; ESV, end-systolic volume; FS, fractional shortening; LVDd, left ventricular end-diastolic dimension; LVDs, left ventricular end-contractile diameter; HW, heart weight; BW, body weight; TL, tibia length; WGA, wheat germ agglutinin; CM, cardiomyocyte; SHAM, sham surgery; TAC-SE, transverse aortic constriction-sedentary; and TAC-EX, transverse aortic constriction-exercise.

**Figure 2 ijms-23-11189-f002:**
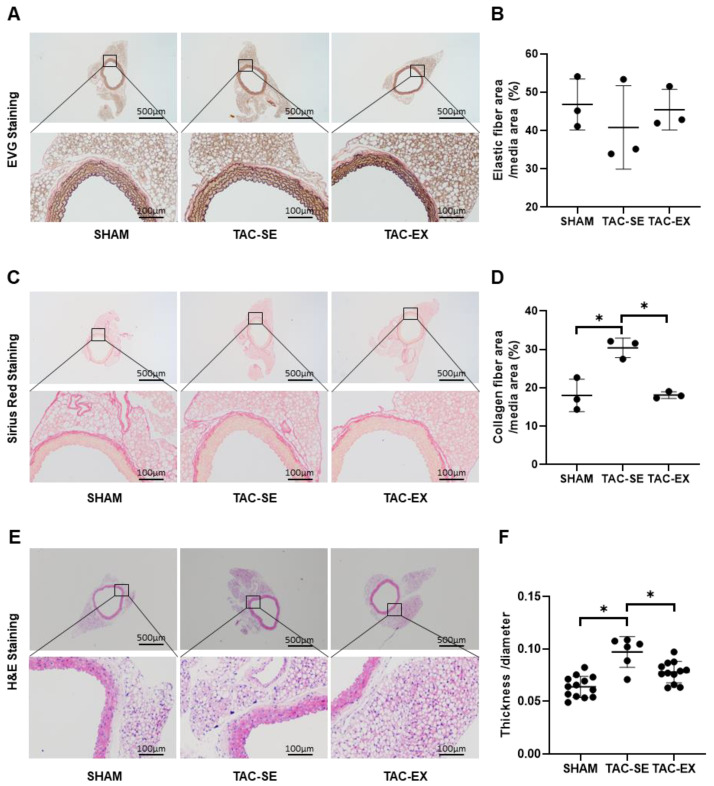
Effects of exercise on aortic elastic/collagen fiber area and aortic morphology in mice after TAC. Elastic fibers of aorta were stained with EVG (**A**,**B**) and collagen fibers with Sirius Red (**C**,**D**) in PVAT. (**E**) Aorta were stained with H&E to compare differences between groups on thickness/diameter (**F**). Data were analyzed using one-way ANOVA; values are mean ± SD. * indicates *p* < 0.05. Abbreviations: EVG, Elastica van Gieson; H&E, hematoxylin and eosin; SHAM, sham surgery; TAC-SE, transverse aortic constriction-sedentary; and TAC-EX, transverse aortic constriction-exercise.

**Figure 3 ijms-23-11189-f003:**
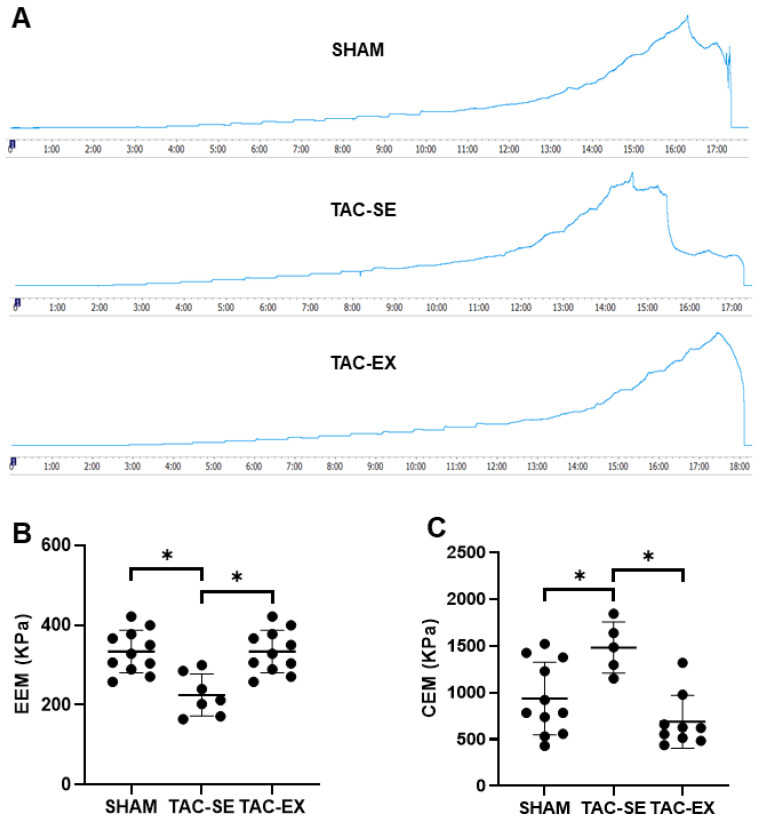
Effects of exercise on aortic stiffness in mice after TAC. Representative images of the force–time curve for each group (**A**); EEM (**B**) and CEM (**C**) were calculated using the slope of stress–strain curves. Data were analyzed using one-way ANOVA; values are mean ± SD. * indicates *p* < 0.05. Abbreviations: EEM, elastin elastic modulus; CEM, collagen elastic modulus; SHAM, sham surgery; TAC-SE, transverse aortic constriction-sedentary; and TAC-EX, transverse aortic constriction-exercise.

**Figure 4 ijms-23-11189-f004:**
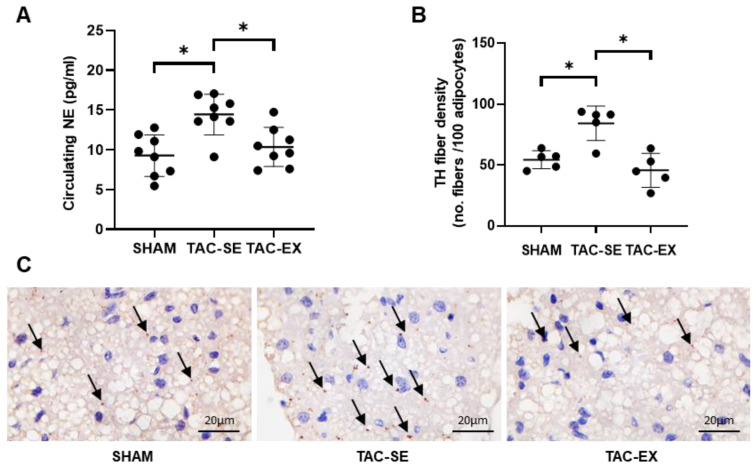
Effects of exercise on hyperexcitation of sympathetic nerves in PVAT of mice after TAC. Circulating levels of NE were determined using ELISA (**A**). Density of TH-immunoreactive parenchymal nerve fibers were calculated by the number of fibers every 100 adipocytes (**B**). Sympathetic nerve fibers were stained with TH antibody in PVAT, as indicated by arrowheads (**C**). Data were analyzed using one-way ANOVA; values are mean ± SD. * indicates *p* < 0.05. Abbreviations: NE, noradrenaline; TH, tyrosine hydroxylase; SHAM, sham surgery; TAC, transverse aortic constriction; TAC-SE, transverse aortic constriction-sedentary; and TAC-EX, transverse aortic constriction-exercise.

**Figure 5 ijms-23-11189-f005:**
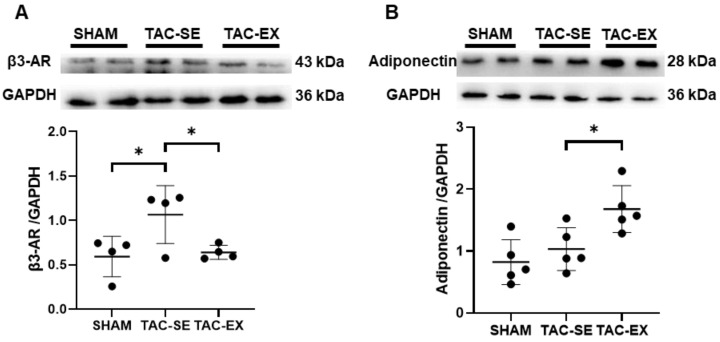
Effects of exercise on the expressions of β3-AR and adiponectin in PVAT of mice after TAC. Representative Western blot assessment of β3-AR (**A**) and adiponectin (**B**) expressions in PVAT normalized to the expressions of GAPDH. Data were analyzed using one-way ANOVA; values are mean ± SD. * indicates *p* < 0.05. Abbreviations: β3-AR, β3-adrenergic receptor; SHAM, sham surgery; TAC-SE, transverse aortic constriction-sedentary; and TAC-EX, transverse aortic constriction-exercise.

**Table 1 ijms-23-11189-t001:** Effect of exercise on general characteristics of experimental animals after TAC.

	SHAM (*n* = 15)	TAC-SE (*n* = 12)	TAC-EX (*n* = 15)
Body Weight (g)	27.43 ± 2.09	26.65 ± 1.69	26.78 ± 1.90
PVAT (mg)	8.68 ± 1.83	7.31 ± 1.99	10.02 ± 2.29 *
PVAT (mg)/Body Weight (g)	0.318 ± 0.064	0.269 ± 0.076	0.375 ± 0.071 *

Abbreviations: PVAT, perivascular adipose tissue; SHAM, sham surgery; TAC-SE, transverse aortic constriction-sedentary; and TAC-EX, transverse aortic constriction-exercise. Data were analyzed using one-way ANOVA; values are mean ± SD. * indicates *p* < 0.05 vs. TAC-SE.

## Data Availability

The data presented in this study are available on request from the corresponding authors.

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
