# Peer review of "Aerobic Exercise Prevents Arterial Stiffness and Attenuates Hyperexcitation of Sympathetic Nerves in Perivascular Adipose Tissue of Mice after Transverse Aortic Constriction"

_ijms, 2022, doi:10.3390/ijms231911189_

Round 1

Reviewer 1 Report

Research article title: Aerobic Exercise Prevents Arterial Stiffness and Attenuates Hy- 2 perexcitation of Sympathetic Nerves in Perivascular Adipose 3 Tissue of Mice after Transverse Aortic Constriction

 The submitted article needs a minor revision in order to bring better quality of scientific presentation towards the easy understanding of the article. The authors are advised to respond thoroughly to all the corrections mentioned below, which are absolutely not limited but applicable to the entire manuscript. I recommend this manuscript for publication, once the authors consider the revisions for the below comments.

Thanks for considering my review.

1. A few places where references for important points that substantiate the logic or discussion of results are lacking.

2. Include statistical tool name in the method section.

3. Authors should explain the certain time frame for aerobic exercise used in this study

4. Authors should provide more detailed introduction part.

5. Author should check typographical error all over the manuscript.

Reviewer 2 Report

Dear authors,

I have studied with great interest the manuscript « Aerobic Exercise Prevents Arterial Stiffness and Attenuates Hyperexcitation of Sympathetic Nerves in Perivascular Adipose Tissue of Mice after Transverse Aortic Constriction».

The work presented is original and may have potential relevance in the future for the clinical management of patients after aortic constriction. The authors hypothesized that Regular aerobic exercise would effectively prevent arterial stiffness and extracellular matrix remodeling in the developmental course of HF, during which sympathetic innervation and adiponectin within PVAT might be strongly implicated. Results are clearly exposed and well written. However, I must mind some other details that I guess will improve the quality of the paper:

1.       The authors should add a couple of world about the role of adiponectin in the introduction to highlight why you investigated this biomarker in your study.

2.       It a little bit strange that section “Materials and Methods followed by Discussion». I suppose it would be better to replace it after Introduction.   
